# HiddenKey: Parameter-Efficient FineTuning Meets Dropout under a Unified Framework

## Abstract

The emerging powerful capabilities exhibited by large language models (LLMs) have established them as a fundamental element in various applications that rely on advanced language understanding. At the same time, fine-tuning has become the standard learning approach to adapting LLMs to a concrete application (e.g., instruction tuning, alignment tuning, and task/user-specific specialization). Due to the high cost associated with full finetuning, parameter-efficient finetuning (PEFT) methods, especially LoRA, have gained popularity due to their lower storage, memory, and computation requirements. However, the possible contradiction between limited trainable parameters and the dropout regularization methods (which aim at alleviating overfitting associated with excessive parameter redundancy), has been largely overlooked. With extensive experiments of LoRA-based PEFT, we first confirm that PEFT is also overfitting-prone. We then revisit transformer-specific dropout methods, and validate their equivalence and differences mathematically and empirically. To facilitate a comprehensive comparison, we introduce a unified framework to instantiate them along dropping position, structural pattern and compensation measure, and uncover their new preferences and performance comparisons in PEFT scenarios. This framework also enables us to integrate the best of all into a new dropout method named HiddenKey, which shows performance superiority over existing methods on both NLU and NLG tasks. Compared to baselines, it also achieves better performance with less fine-tuning time, and offers continuous improvement with further finetuning. These highlight HiddenKey as the better practice for high-performance and parameter-efficient finetuning of LLMs.

## 1 Introduction

With efficient architecture, good scalability and outstanding performance, transformers have gradually dominated sequence modeling in natural language processing (NLP) and have been widely applied to the fields of image and speech (Vaswani et al., 2017; Kirillov et al., 2023; Radford et al., 2023). In general, a model's performance tends to improve with more parameters given sufficient training data. Thus, pretrained language models, such as GPT-4 (OpenAI, 2023), PaLM 2 (Anil et al., 2023) and LLaMA 2(Touvron et al., 2023b), have been rapidly expanded to hundreds of billions or even trillions of parameters, leading to significant performance improvement. When customizing these models for downstream tasks, fine-tuning has been the standard learning approach. However, in addition to being computationally expensive, full fine-tuning in the increasing multi-task and multi-user scenarios requires storing a complete set of parameters for each user/task, making it less practical due to the high costs associated with storage, training and inference. Hence, many parameter-efficient fine-tuning (PEFT) approaches have been proposed (Houlsby et al., 2019; Hu et al., 2021; Liu et al., 2022). These methods freeze the vast majority of parameters, allowing or adding only a small portion of parameters to be updated. Performance comparable to or better than fine-tuning a whole model is reported, with less than 0.1% of the model parameters adjusted (Hu et al., 2021; He et al., 2021). Meanwhile, PEFT methods have been proven to be less prone to overfitting and enjoy better robustness (Chen et al., 2022). PEFT methods, especially LoRA, have hence been widely studied in academia and deployed in industry (Valipour et al., 2022).

However, PEFT and dropout present a possible contradiction when jointly applied: the dropout methods can be ineffective with limited trainable parameters in PEFT scenarios, as overfitting

mostly occurs with excessive parameter redundancy. In this study, we first conduct extensive experiments with LoRA-based PEFT, and confirm that PEFT also suffers from overfitting easily and can be improved with dropout methods, regardless of the inconsistent choices of dropout methods between pretraining and finetuning stages. Besides, we analyze existing transformer-specific dropout methods mathematically, and find the quantitative relationship between DropKey and DropAttention Empirically, these methods have new preferences in PEFT scenarios. For example, span-wise HiddenCut is no longer superior to the element-wise one due to the limited tunable parameters, while DropKey prefers the column style for NLP tasks instead of the element style for computer vision (CV) tasks.

To compare these methods comprehensively, we introduce a unified framework from the perspective of dropout positions, structural pattern and compensation measure. Within this framework, we find that DropKey performs the best followed by HiddenCut, and DropAttention exhibits the worst performance due to the unreasonable gradient noise. Bidirectional Kullback-Leibler (KL) divergence loss consistently achieves performance gains, while Jensen-Shannon (JS) consistency regularization loss does not. Guided by this framework to integrate the best of all, we also derive a new dropout method named HiddenKey. HiddenKey empirically exhibits superiority on multiple natural language understanding (NLU) and natural language generation (NLG) tasks, filling the gap on the effect of dropout methods on NLG tasks largely overlooked by previous research. Integrating with input and output dropout does not provide further consistent complementarity, demonstrating the adequacy of our method. It also outperforms baselines with less finetuning time, and continuous performance improvement can be obtained when longer finetuning process is allowed. Hence, HiddenKey excels as the better method for high-performance and parameter-efficient finetuning of LLMs on both NLU and NLG tasks.

## 2 PRELIMINARIES

We revisit three transformer-specific dropout methods shown in Figure 1 and use the mathematical symbols for the analysis of their features and interconnections in the next section.

**DropAttention.** DropAttention (Zehui et al., 2019) is the first dropout-based regularization method specially designed for self-attention mechanism. It randomly masks independent elements or key columns of attention weights, to encourage the model to utilize different contextualized features instead of overfitting some specific patterns. Following Eq. 1 and Eq. 2, normalized rescaling operation replaces traditional rescaling to guarantee that the sum of attention weights remains 1, and achieves better performance and more stable training for multiple NLP classification tasks.

$$\overline{w}_j = m \cdot w_j, \quad m \sim \text{Bernoulli}(p) \tag{1}$$

$$w'_j = \frac{\overline{w}_j}{\text{NoGrad}(\sum_{j=0}^{l-1} \overline{w}_j)} \tag{2}$$

where $p$, $w_j$, $\overline{w}_j$ and $w'_j$ are the dropout rate, original, masked and rescaled attention weights, while $\text{NoGrad}()$ and $\text{Bernoulli}()$ denote gradient stopping operator and sampling from Bernoulli distribution, respectively[1].

**DropKey.** Instead of dropping attention weights after the $\text{softmax}()$ operation, DropKey (Li et al., 2023) proposes a dropout-before-softmax scheme which takes attention logits $g_j$ as the dropout units, shown in Eq. 3. Since the subsequent $\text{softmax}()$ ensures the sum of weights to be one, rescaling is unnecessary in DropKey. Although three structured dropout methods are introduced, the vanilla structure is verified to be the most effective for CV tasks (Li et al., 2023).

$$g'_j = m + g_j, \quad m = \begin{cases} 0, & \text{with probability } 1-p \\ -\infty, & \text{with probability } p \end{cases} \tag{3}$$

---

[1] Here we omit the subscript $t$ for clarity. Although whether the $\text{NoGrad}()$ operator exists or not significantly impacts the performance of DropAttention, it is overlooked in the original paper. We use it here and will discuss both cases in detail.

**HiddenCut.** In contrast, HiddenCut (Chen et al., 2021) focuses on preventing the co-adaptation of hidden representations produced in the feed-forward module. The core idea is to cut single contiguous span, which may contain semantic information that is more difficult to be restored than independent hidden elements. Additionally, JS loss is applied to encourage the perturbed representations to be as close to the ones in inference as possible.

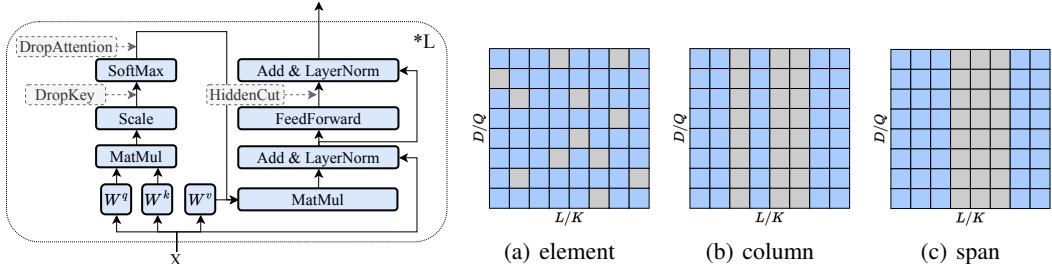

Figure 1: Illustration of transformer architecture and popular transformer-specific dropout methods, namely DropKey, DropAttention and HiddenCut. Blocks with dashed borderlines and arrows represent positions of these methods.

Figure 2: Three structural sampling strategies, namely element, column and span. The grey and blue cells represent masked and remaining entries, respectively. Rows and columns represent the sequence length and the hidden state dimension for HiddenCut, and the numbers of keys and queries for DropKey and DropAttention.

## 3 METHOD

In this section, we first conduct an analytical study of existing dropout methods. We then propose a unified framework from the perspective of dropout positions, structural pattern and compensation measure. Finally, we derive a new dropout method named HiddenKey which provides respective treatment to attention logits and feed-forward module and exhibits better performance empirically.

### 3.1 MATHEMATICAL AND EMPIRICAL COMPARISON

**Equivalent Forwarding between DropKey and DropAttention.** Even if the dropping details are different between DropKey and DropAttention, we show their mathematical equivalence in the forward pass. Let $g_u$ and $g_m$ represent the unmasked and masked attention logits, and $w_u$ and $w_m$ denote attention weights[2]. For DropKey, we have

$$g'_m := -\infty, \quad g'_u := g_u, \quad w'_m = 0 \tag{4}$$

$$w'_u = \frac{\exp(g'_u)}{\sum_{i=0}^{l-1} \exp(g'_i)}, \tag{5}$$

while for DropAttention, we have

$$w'_m := 0 \tag{6}$$

$$w'_u = \frac{\exp(g_u)}{\sum_{i=0}^{l-1} \exp(g_i)} \cdot \frac{1}{\sum_{i=0}^{l-1} \overline{w}_i} \tag{7}$$

Proved by Eq. 14 in Appendix B, Eq. 5 and Eq. 7 are strictly equal to each other. Hence, the final attention weights (i.e., $w'_u$ and $w'_m$) of DropKey are the same as those of DropAttention in the forward pass, and so is the following computation in the whole model. It is worth noting that normalized rescaling plays an important role in this equivalence, which diminishes the differences between these two methods in the forward process.

---

[2]Only one masked element is considered here, but masking multiple elements shares the same analysis.

**Similarity of Back-Propagation between DropKey and DropAttention.** Given $L$ as the loss function, the corresponding values of $\frac{\partial L}{\partial w'_u}$ and $\frac{\partial L}{\partial w'_m}$ remain the same for DropKey and DropAttention due to the identical forward pass. Meanwhile, because the forward pass before calculating the attention logits is also the same, the analysis of back-propagation focuses on the four partial derivatives of $w'_u$ and $w'_m$ with respect to $g_u$ and $g_m$, respectively. For DropKey, we have

$$\frac{\partial w'_u}{\partial g_u} = \exp(g_u) \cdot \frac{\sum_{i=0,\neq m}^{l-1} \exp(g_i) - \exp(g_u)}{(\sum_{i=0,\neq m}^{l-1} \exp(g_i))^2}. \tag{8}$$

For DropAttention with $\mathrm{NoGrad}()$ operator, we have

$$\frac{\partial w'_u}{\partial g_m} = -\frac{\exp(g_u) \cdot \exp(g_m)}{\sum_{i=0}^{l-1} \exp(g_i) \cdot \sum_{i=0,\neq m}^{l-1} \exp(g_i)} \tag{9}$$

$$\frac{\partial w'_u}{\partial g_u} = \frac{\exp(g_u) \cdot \sum_{i=0,\neq u}^{l-1} \exp(g_i)}{\sum_{i=0}^{l-1} \exp(g_i) \cdot \sum_{i=0,\neq m}^{l-1} \exp(g_i)} \tag{10}$$

For other partial derivatives, the dropping operations stop the gradient flow or set the gradients to 0. When the corresponding elements of attention logits and weights are masked, the derivative of $w'_u$ with respect to $g_u$ has proportional relation as shown in Eq. 11, proven in Eq. 15 of Appendix B. Provably, $k$ is always less than 1 and continuously decreases with the increase of $g_m$. In other words, compared to DropAttention with $\mathrm{NoGrad}()$ operator, DropKey can adaptively lower the gradients when a large attention logit $g_m$ is discarded. This can provide DropKey with dropping-dependent compensation capability, thereby stabilizing the training process. For DropAttention with $\mathrm{NoGrad}()$, the partial derivative of $w'_u$ with respect to $g_m$ is none-zero and that with respect to $g_u$ depends on the value of $g_m$, even if $w_m$ is masked and $g_m$ does not participate in the forward computation. This implies that a larger dropout rate can introduce more gradient noise. The inferior performance in Sec. 4 also validates this analysis empirically. In contrast, DropAttention without $\mathrm{NoGrad}()$ provably shares the same back-propagation with HiddenKey, thereby exhibiting identical behaviors. Hence, unless otherwise stated, we will refer to DropAttention with $\mathrm{NoGrad}()$ as DropAttention, and include DropAttention without $\mathrm{NoGrad}()$ under DropKey for simplicity below.

$$(\frac{\partial w'_u}{\partial g_u})^{\mathrm{DropKey}} = k \cdot (\frac{\partial w'_u}{\partial g_u})^{\mathrm{DropAttention}}, \quad k = \frac{1 - \frac{\exp(g_u)}{\sum_{i=0,\neq m}^{l-1} \exp(g_i)}}{1 - \frac{\exp(g_u)}{\sum_{i=0}^{l-1} \exp(g_i)}} \tag{11}$$

**Comparison with HiddenCut.** The commonality among these methods is that they all need to select a specific type of data, decide what patterns to mask, and consider how to reduce the differences between the training and inference stages, which will be explained in detail in the next section. In contrast, the differences are two-fold. First, their distinct dropping locations and patterns leads to different compensation methods. Similar to the vanilla dropout, element-wise HiddenCut amplifies representations by a factor of $1/(1-p)$ for consistent scales between training and testing, while normalized rescaling in Eq. 2 is used for DropAttention. Due to the subsequent $\mathrm{softmax}()$, DropKey no longer uses any scaling method. Next, the other difference is that like DropConnect (Wan et al., 2013) which randomly zeros layer weights, DropAttention and DropKey can also be regarded as operations performed on weight matrices (which are utilized for the weighted summation of value vectors), even if these weight matrices are input-dependent and dynamically generated (Zehui et al., 2019). Contrarily, HiddenCut operates directly in semantic representations.

## 3.2 A UNIFIED FRAMEWORK

Although these methods has different design details, the core idea and components behind designing a dropout method are shared, such as dropping position, structural pattern and compensation strategy to close the training and inference gap. We will further explain these design dimensions and instantiate these dropout methods along them below.

**Dropping Position.** For better generalization, a robust model needs to learn noise-resilient features. Hence, dropping position, determining where to inject noise, emerges as a primary consideration in designing dropout methods. For example, dropping inputs acts like data augmentation,

dropping output representations encourages an ensemble of sub-classifiers, and dropping intermediate representations affects the co-adaptation of different neurons. For a transformer layer illustrated in Figure 1, DropKey, DropAttention and HiddenCut respectively drop attention logits, weights and hidden representations, covering the self-attention mechanism and feed-forward module.

**Structural Pattern.** Structural pattern means the style of units deactivated randomly, and determines how the co-adaptation of neurons is disrupted, thereby affecting the semantic information learned by these units. Specifically, as shown in Figure 2(b), if column pattern is adopted in DropKey, each value vector will tend to possess as much contextual information as possible so that the output vectors is minimally affected by the dropped key columns. Furthermore, different optimal dropping patterns may be required for distinct dropping positions, which will be thoroughly studied.

**Compensation for Training-Inference Gap.** For better performance and deterministic output, dropout mechanisms are generally disabled in inference. However, this is not consistent with the training stage and can lead to a gap between the actual and ideal performance. Therefore, another key consideration is how to close the training and inference gap. Apart from simple rescaling associated with each method, R-drop (Wu et al., 2021) explores another solution which utilizes Eq. 12, bidirectional KL divergence loss, to enforce the output distributions to be more dropout-insensitive so that the gap between training and inference can be minimized. Instead, HiddenCut replaces it with JS loss shown in the Eq. 13. For the sake of symmetry, KL loss calculates the bidirectional distances, while JS loss uses the inference distribution as reference.

$$\mathcal{L}_{KL} = \frac{1}{2}(D_{\mathrm{KL}}(P_1\|P_2) + D_{\mathrm{KL}}(P_2\|P_1)) \tag{12}$$

$$\mathcal{L}_{JS} = D_{\mathrm{KL}}(P_1\|\overline{P}) \tag{13}$$

where $P_1$, $P_2$ and $\overline{P}$ represent two different output distributions in the training stage and one in inference with the same samples, respectively. $D_{\mathrm{KL}}()$ calculates the asymmetric KL divergence between two distribution.

### 3.3 HIDDENKEY

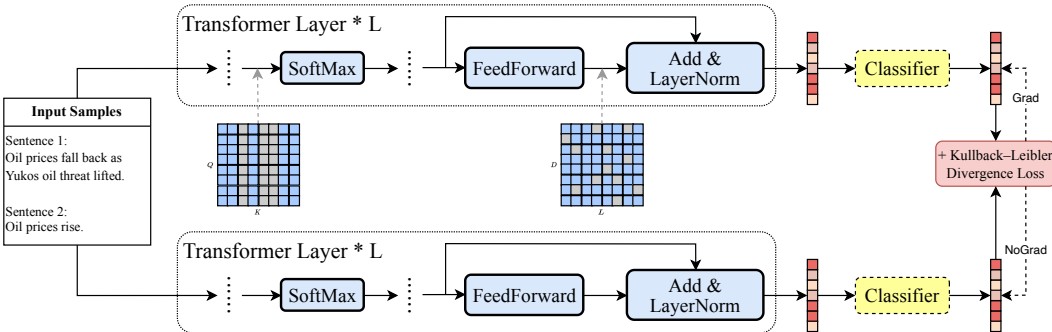

Figure 3: Illustration of DropKey. It respectively drops columns and elements for attention logits and hidden representations, and augments KL loss to minimize the training and testing gap implicitly.

The proposed unified framework enables us to analyze the critical choices along each dimension and their mutual influences, and also guides us to design new dropout methods. As shown in Figure 3, we propose "HiddenKey", which drops the attention logits column-wisely in the attention mechanism and hidden representations element-wisely in the feed-forward module. Similar to a Siamese architecture (Bromley et al., 1993), during the training stage, the model performs two forward passes in parallel which does not apparently increase the training time, and augments an extra KL loss to enhance the similarity of output distributions, thereby minimizing the gap between training and testing. For classification tasks, the representations produced by the classifier are used, while those produced by the last transformer layer are used for regression tasks. Furthermore, the superiority over all the aforementioned methods will be extensively analyzed on diverse tasks and models below.

## 4 EXPERIMENTS

### 4.1 GENERAL SETUP

**Models and Datasets.** We implement comprehensive empirical analysis of multiple tasks and models in LoRA-based PEFT scenarios. The models start from RoBERTa-large (Liu et al., 2019) and GPT2-Medium (Li & Liang, 2021) and scale up to LLaMA-7B(Touvron et al., 2023a). The tasks cover NLU and NLG. For NLU tasks, we use six datasets from the GLUE benchmark (Wang et al., 2018): (i) **SST-2** (Socher et al., 2013), (ii) **RTE** (Wang et al., 2018), (iii) **MRPC** (Dolan & Brockett, 2005), (iv) **STS-B** (Cer et al., 2017), (v) **CoLA** (Warstadt et al., 2018) and (vi) **QNLI** (Rajpurkar et al., 2018). These datasets are selected to cover different sizes and diverse tasks, including single sentence, similarity, paraphrase and inference. Specially, STS-B performs a regression task for better generalization of our conclusions. For NLG tasks, we follow Hu et al. (2021) and focus on **E2E** (Novikova et al., 2017) and **WebNLG** (Gardent et al., 2017). More details can be found in Appendix C.

**Baseline.** Due to the popularity, we use models with LoRA as the representative baselines for PEFT scenarios, and keep their most configurations. Especially, parallel low-rank matrices are utilized with the rank as 8 and a scaler as 16 for $W^k$ and $W^v$ in the attention module. This results in trainable parameters of 0.79M in the Roberta-large model, accounting for 0.22% of the total model parameters[3]. In comparison, for GPT2-Medium, these values are 0.39M and 0.11%, while for LLaMA-7B, they are 4.19M and 0.06%. The detailed configurations are in the Appendix D.

### 4.2 MAIN RESULTS

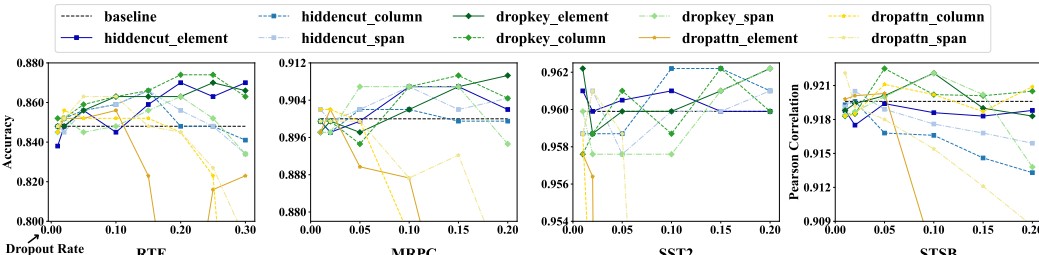

Figure 4: Performance of different dropout methods on four datasets (RTE, MRPC, STS-B and SST-2). Markers and line styles differentiate various dropping positions, while the shades of color represent the structural patterns. Pearson correlation is reported for STS-B, and accuracy for others.

In the PEFT scenario, we first conduct extensive experiments with RoBERTa-large on four NLU datasets, and display the results in Figure 4 and Table 1. Generally, almost all methods can outperform the baseline with a large margin. This demonstrates that despite limited trainable parameters, PEFT still suffers from overfitting and these transformer-specific dropout methods can alleviate this problem regardless of different dropout methods from the pretraining stage. We claim that limited adjustable parameters of PEFT in LLMs still enable large model capacity. This stems from two aspects: (1) Even if the proportion is negligible, the number of tunable parameters remains significant due to the large size of foundation models. In our setting, there are still 0.79M tunable parameters, accounting for 0.22% of the total in Roberta-large model. (2) Coupled with the base model, the expressiveness of these parameters is enlarged extremely, as evidenced by the remarkable performance across multiple tasks in (Hu et al., 2021; Zaken et al., 2021). This excessive model capacity causes the entire model to overfit to specific patterns, with a negligible portion of tunable parameters.

Different dropout methods have distinctive characteristics in PEFT scenarios. Specifically, with a small dropout rate, all methods perform very similarly, fluctuating around the baseline. However, as the dropout rate increases, DropKey consistently achieves the best performance on four datasets, followed by HiddenCut. Both of them also exhibit an overall trend of initially increasing and then

---

[3]The classifier parameters are excluded here due to their varying numbers for different tasks.

decreasing, which aligns with the regular dropout behaviors in pre-training and full-finetuning (Srivastava et al., 2014). In contrast, despite the similar dropping positions and the same forward pass as DropKey, DropAttention produces the worst results. This confirms our earlier analysis in Sec. 3.1, indicating that the backward propagation with $\mathrm{NoGrad}()$ leads to larger gradient noise and rapidly degrades the model's performance as the dropout rate increases.

Table 1: Results of various dropping positions, structural patterns and compensation methods for RoBERTa-large model on RTE, MRPC, STS-B and SST-2 datasets. "input" and "output" mean the dropping of input and output representations. The superscript of data denotes the optimal parameter of each method, while the subscript means the standard deviation. Bold indicates the best performance. "Compen." and "Avg." are compensation methods and the average results on four datasets. [4]

| Position | Pattern / Compen. | RTE | MRPC | STS-B | STS2 | Avg. |
| --- | --- | --- | --- | --- | --- | --- |
| | | Acc. | Acc. | Pearson. | Acc. | |
| baseline | - | $84.48^{0.00}_{\pm0.98}$ | $89.95^{0.00}_{\pm0.50}$ | $91.96^{0.00}_{\pm0.48}$ | $95.99^{0.00}_{\pm0.25}$ | 90.60 |
| HiddenCut | element | $87.00^{0.20}_{\pm1.14}$ | $90.69^{0.10}_{\pm0.42}$ | $91.94^{0.05}_{\pm0.28}$ | $96.10^{0.10}_{\pm0.42}$ | 91.43 |
| | column | $86.64^{0.15}_{\pm0.80}$ | $90.20^{0.05}_{\pm0.80}$ | $91.96^{0.02}_{\pm0.11}$ | $96.22^{0.10}_{\pm0.19}$ | 91.26 |
| | span | $86.64^{0.15}_{\pm1.63}$ | $90.69^{0.10}_{\pm0.22}$ | $92.05^{0.02}_{\pm0.35}$ | $96.10^{0.20}_{\pm0.30}$ | 91.37 |
| DropKey | element | $87.00^{0.25}_{\pm1.08}$ | $90.93^{0.20}_{\pm1.06}$ | $92.21^{0.10}_{\pm0.21}$ | $96.22^{0.20}_{\pm0.25}$ | 91.59 |
| | column | $87.36^{0.20}_{\pm1.70}$ | $90.93^{0.15}_{\pm0.40}$ | $92.25^{0.05}_{\pm0.13}$ | $96.22^{0.15}_{\pm0.24}$ | 91.69 |
| | span | $86.28^{0.20}_{\pm0.94}$ | $90.69^{0.05}_{\pm0.69}$ | $92.21^{0.10}_{\pm0.21}$ | $96.22^{0.20}_{\pm0.25}$ | 91.35 |
| DropAttention | element | $85.56^{0.10}_{\pm11.73}$ | $90.20^{0.02}_{\pm3.07}$ | $92.03^{0.05}_{\pm0.27}$ | $95.76^{0.01}_{\pm0.30}$ | 90.89 |
| | column | $85.56^{0.02}_{\pm1.80}$ | $90.20^{0.02}_{\pm0.71}$ | $92.11^{0.05}_{\pm0.28}$ | $95.87^{0.01}_{\pm0.21}$ | 90.94 |
| | span | $86.28^{0.05}_{\pm0.60}$ | $89.95^{0.01}_{\pm0.61}$ | $92.21^{0.01}_{\pm0.36}$ | $96.10^{0.02}_{\pm0.39}$ | 91.14 |
| HiddenKey$^-$ | - | $87.70^{0.05,0.2}_{\pm0.91}$ | $90.90^{0.05,0.15}_{\pm0.72}$ | $92.28^{0.05,0.05}_{\pm0.19}$ | $96.22^{0.1,0.15}_{\pm0.13}$ | 91.78 |
| | + KL | $88.10^{0.50}_{\pm1.60}$ | $\mathbf{91.20}^{5.00}_{\pm0.90}$ | $\mathbf{92.30}^{1000.00}_{\pm0.11}$ | $\mathbf{96.44}^{2.00}_{\pm0.20}$ | **92.01** |
| | + JS | $87.70^{0.01}_{\pm1.72}$ | $90.90^{0.02}_{\pm0.47}$ | $92.24^{0.05}_{\pm0.21}$ | $96.22^{0.01}_{\pm0.24}$ | 91.77 |
| | + input | $\mathbf{88.50}^{0.02}_{\pm2.11}$ | $90.70^{0.10}_{\pm1.03}$ | $92.11^{0.05}_{\pm0.14}$ | $96.33^{0.20}_{\pm0.27}$ | 91.16 |
| | + output | $87.70^{0.02}_{\pm2.24}$ | $90.70^{0.10}_{\pm1.20}$ | $92.19^{0.10}_{\pm0.11}$ | $96.22^{0.10}_{\pm0.15}$ | 90.95 |

Dropping positions prefer different optimal structural patterns, and combining different positions can further improve performance. Based on our results, the optimal structure for DropKey is "column", which deactivates specific keys across all queries within a head, thereby breaking the co-adaptation of value vectors and achieving better performance. Oppositely, Li et al. (2023) confirms the ineffectiveness of structural patterns in multiple CV tasks. This divergence may arise from that NLP tasks have a more semantically explicit token segmentation, while this property is absent for CV tasks. In comparison, HiddenCut only has one representation sequence instead of multiple sequences in the multi-head attention module in DropKey. Hence, "column" and "span" modes erase too much information, especially when semantically important representations, such as negation words, emotional words, etc., are masked. This may introduce excessive noise and even incorrect input-label pairs for the relatively limited PEFT capacity, and explains why element-wise HiddenCut achieves better performance on average, contrary to the superiority of "span" in full-finetuning (Chen et al., 2021). In Table 1, we further combine element-wise HiddenCut with column-wise DropKey, named HiddenKey$^-$. On average, it achieves further improvement compared to any single dropout mechanism. We also attempt to combine DropAttention, but it does not lead to any improvement.

As for the augmented loss to narrow the gap between training and inference, KL loss empirically achieves better performance than JS loss. Specifically, compared to HiddenKey$^-$ (i.e. HiddenKey

---

[4]Even if there are some extra designs among existing dropout methods, like scheduled dropout ratio and strategic sampling, they cannot be shared or are complicated to optimize, thereby being neglected here.

without any additional loss), the introduction of KL loss always provides further performance gains on all datasets, including the STS-B dataset of a regression task. In contrast, JS loss does not have an apparent impact on the results, while Chen et al. (2021) claims its superiority but does not implement any comparison with KL loss. This difference may arise from the PEFT scenario and more superb dropout method, which squeezes the potential improvement space of JS loss. Therefore, with the validated superior performance, KL loss is applied in HiddenKey along the third dimension of our proposed framework. Due to the optimal practice along each dimension, HiddenKey also achieves the best performance among all above methods on all datasets.

## 4.3 GENERALIZATION ON MORE NLU DATASETS AND LLMS

**More NLU Datasets.** We generalize HiddenKey to two extra NLU datasets, namely CoLA and QNLI. As shown in Table. 2, HiddenKey steadily achieves 1.95 and 0.81 performance improvement over baselines on both of the datasets, even if Matthew's correlation is used for the evaluation of CoLA. These further confirm the superiority of HiddenKey in NLU tasks.

Table 2: Results of RoBERTa-large finetuned with HiddenKey on CoLA and QNLI datasets.

| Method | CoLA Matthew. | QNLI Acc. |
|---|---|---|
| baseline | $67.96_{\pm0.25}$ | $94.23_{\pm0.17}$ |
| HiddenKey | $\mathbf{69.91}_{\pm0.52}$ | $\mathbf{95.04}_{\pm0.11}$ |

Table 3: Results of LLaMA-7B finetuned with HiddenKey on RTE and MRPC datasets.

| Method | RTE Acc. | MRPC Acc. |
|---|---|---|
| baseline | $89.17_{\pm1.18}$ | $88.73_{\pm0.62}$ |
| HiddenKey | $\mathbf{90.61}_{\pm1.51}$ | $\mathbf{89.22}_{\pm1.12}$ |

**LLaMA.** With the dominance of LLMs, we also extend the application of HiddenKey to LLaMA 7B, one of the most popular and open-sourced LLMs. Its performance on RTE and MRPC datasets is given in Table 3. Apparently, models finetuned with HiddenKey outperform those without HiddenKey by a large margin in the LoRA-based PEFT scenario. This indicates that HiddenKey can also function well when an auto-regressive decoder-only LLM is deployed and the representation of the final time step is used for prediction.

## 4.4 PERFORMANCE ON NLG TASKS

Following Hu et al. (2021), we conduct extra experiments with GPT2-Medium to demonstrate the superior performance of HiddenKey on NLG tasks. As shown in Table 4, HiddenKey consistently outperforms baseline and other dropout methods over all the five metrics on E2E dataset. Similarly in Table 5, on the "All", "Seen" and "Unseen" subsets of the WebNLG dataset, HiddenKey gains 7/9 wins over other methods on BLEU, METEOR and TER metrics. In conclusion, HiddenKey exhibits similar performance surge on diverse metrics, datasets and their subsets with causal autoregressive models for NLG tasks as it has shown for NLU tasks, and can be the recommended method for high-performance and parameter-efficient finetuning of LLMs on both NLU and NLG tasks.

Table 4: Results of GPT2-Medium finetuned with different dropout methods on E2E dataset.

| Method | BLEU ↑ | NIST ↑ | METEOR ↑ | ROUGE_L ↑ | CIDEr ↑ |
|---|---|---|---|---|---|
| Baseline | $68.50_{\pm0.90}$ | $8.615_{\pm0.09}$ | $0.464_{\pm0.00}$ | $0.711_{\pm0.00}$ | $2.490_{\pm0.02}$ |
| HiddenCut | $69.22_{\pm0.44}$ | $8.700_{\pm0.05}$ | $0.467_{\pm0.00}$ | $0.714_{\pm0.00}$ | $2.491_{\pm0.01}$ |
| DropKey | $68.78_{\pm0.75}$ | $8.651_{\pm0.08}$ | $0.465_{\pm0.00}$ | $0.714_{\pm0.00}$ | $2.486_{\pm0.01}$ |
| HiddenKey$^-$ | $69.35_{\pm0.48}$ | $8.726_{\pm0.04}$ | $0.466_{\pm0.00}$ | $0.716_{\pm0.00}$ | $2.510_{\pm0.00}$ |
| HiddenKey | $\mathbf{69.76}_{\pm0.51}$ | $\mathbf{8.765}_{\pm0.08}$ | $\mathbf{0.468}_{\pm0.00}$ | $\mathbf{0.718}_{\pm0.00}$ | $\mathbf{2.511}_{\pm0.03}$ |

## 4.5 COMPLEMENTARITY WITH INPUT AND OUTPUT DROPPING

Except various dropout methods designed for transformer layers, cutoff, proposed in Shen et al. (2020), is applied to input embedding sequence to perform data augmentation, and standard dropout

Table 5: Results of GPT2-Medium finetuned with different dropout methods on WebNLG dataset. "A", "S" and "U" denote the "All", "Seen" and "Unseen" categories in the test set, correspondingly.

| Method | A | | | S | | | U | | |
|---|---|---|---|---|---|---|---|---|---|
| | BLEU ↑ | METEOR ↑ | TER ↓ | BLEU ↑ | METEOR ↑ | TER ↓ | BLEU ↑ | METEOR ↑ | TER ↓ |
| Baseline | $54.78_{\pm 0.16}$ | $0.411_{\pm 0.00}$ | $0.395_{\pm 0.00}$ | $62.30_{\pm 0.47}$ | $0.420_{\pm 0.04}$ | $0.331_{\pm 0.00}$ | $45.53_{\pm 0.21}$ | $0.376_{\pm 0.00}$ | $0.464_{\pm 0.00}$ |
| HiddenCut | $55.06_{\pm 0.18}$ | $0.411_{\pm 0.00}$ | $0.391_{\pm 0.00}$ | $62.43_{\pm 0.21}$ | $\mathbf{0.442}_{\pm 0.00}$ | $0.329_{\pm 0.00}$ | $46.11_{\pm 0.20}$ | $0.377_{\pm 0.00}$ | $0.458_{\pm 0.00}$ |
| DropKey | $55.22_{\pm 0.34}$ | $0.411_{\pm 0.00}$ | $0.389_{\pm 0.00}$ | $62.47_{\pm 0.17}$ | $0.441_{\pm 0.00}$ | $0.328_{\pm 0.00}$ | $46.39_{\pm 0.75}$ | $0.378_{\pm 0.00}$ | $0.455_{\pm 0.01}$ |
| HiddenKey$^-$ | $55.26_{\pm 0.20}$ | $0.411_{\pm 0.00}$ | $0.388_{\pm 0.00}$ | $\mathbf{62.57}_{\pm 0.24}$ | $0.441_{\pm 0.00}$ | $0.328_{\pm 0.00}$ | $46.36_{\pm 0.34}$ | $0.378_{\pm 0.00}$ | $0.454_{\pm 0.00}$ |
| HiddenKey | $\mathbf{55.27}_{\pm 0.21}$ | $\mathbf{0.413}_{\pm 0.00}$ | $\mathbf{0.386}_{\pm 0.00}$ | $62.49_{\pm 0.18}$ | $0.441_{\pm 0.00}$ | $\mathbf{0.326}_{\pm 0.00}$ | $\mathbf{46.48}_{\pm 0.46}$ | $\mathbf{0.381}_{\pm 0.00}$ | $\mathbf{0.452}_{\pm 0.00}$ |

is also used to the output representations for a more robust classifier. Therefore, we also investigate whether these methods could further enhance the transformer-specific dropout. The results at the end of Table 1 suggest that neither of these methods consistently achieve improvement over HiddenKey$^-$ across all datasets, and both of their average performance suffer a slight decrease. This indicates that the performance gains brought by dropout methods have mostly been captured by HiddenKey, and the input or output dropping does not provide steady complementarity. This adequacy hints that finetuning with HiddenKey only is enough in PEFT scenarios.

## 4.6 FINETUNING DYNAMICS

Beyond the superior performance on both NLU and NLG tasks, we also visualize the finetuning dynamics to understand HiddenKey from a new perspective. Figure 5 shows the averaged dynamic curves of training loss and evaluation accuracy over five random seeds on RTE dataset, when models are finetuned with different methods in LoRA-based PEFT scenarios. Compared to the baseline whose training loss rapidly converges to near zero, the introduction of HiddenKey$^-$ significantly slows down this process and leads to larger final loss, while HiddenKey further exacerbates this phenomenon. However, large final loss does not mean inferior performance. Specifically, after reaching a fair peak value, accuracy of the baseline deteriorates slightly with the continuous loss decline. This hints that the models suffer from overfitting, which further supports our earlier analysis. In contrast, HiddenKey$^-$ reaches the peak accuracy more slowly with larger loss but remains superior to the baseline. With the

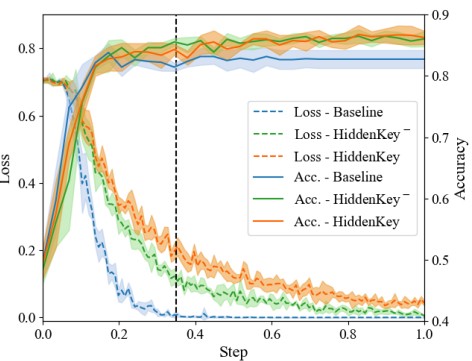

Figure 5: Finetuning loss and evaluation accuracy curves for baseline, HiddenKey$^-$ and HiddenKey. The vertical black dashed line denotes the convergence point of baseline.

additional KL loss, the accuracy keeps fluctuating upwards and achieves the best value, even with the largest loss. It can be anticipated that a longer finetuning process would result in higher accuracy for HiddenKey. In summary, LoRA-based PEFT scenarios are still overfitting-prone and HiddenKey can provide excellent model regularization. Besides, as shown by the vertical black dashed line, although HiddenKey converges slower, it still achieves better performance even before the baseline converges. Therefore, we claim that HiddenKey outperforms the baseline with shorter finetuning process and can continue improving performance when further finetuning is allowed.

## 5 CONCLUSION

We investigate the possible contradiction between the limited trainable parameters in LoRA-based PEFT scenarios and overfitting associated excessive parameter redundancy. After confirming the overfitting-prone property, we give a mathematical analysis of existing dropout methods and introduce a unified framework to compare them empirically, which also guides us to derive a new dropout method, HiddenKey. With its superior performance, adequacy and efficiency on both NLU and NLG datasets, HiddenKey deserves to be the recommended dropout method in PEFT scenarios.

## 6 REPRODUCIBILITY STATEMENT

All of our models are based on open-sourced foundation models, including RoBERTa-large (Liu et al., 2019) and GPT2-Medium (Li & Liang, 2021) and LLaMA-7B(Touvron et al., 2023a). Sufficient details to finetune these models can be found in Sec. 4.1 and Appendix D. Besides, we will also release the code upon publication for publicly available reproducibility with minimal effort.

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

# A  RELATED WORK

Transformers consist of stacked attention-based modules, and demonstrate good performance due to its ability to effectively capture long-range contextual information (Vaswani et al., 2017). Based on a casual multi-layer Transformer decoder for language understanding tasks, Radford et al. (2018) achieves significant gains on training from scratch with a new semi-supervised approach paradigm, combining pretraining an autoregressive language model generatively on unlabelled text and fine-tuning on task-specific data. Subsequently, BERT utilises MLM loss to expand this diagram to bidirectional transformers for natural language understanding tasks (Devlin et al., 2018). These two paradigms have opened up the era of large-scale applications of transformer models like GPT-2 (Radford et al., 2019), Megatron-LM (Shoeybi et al., 2019), and T5 (Raffel et al., 2020), quickly dominating the NLP community and continuously pushing the performance boundaries on multiple tasks. Taking GPT-3 (Brown et al., 2020) with 175B parameters as a representative, large language models emerge strong zero-shot and few-shot learning on many tasks and further sparks a scaling frenzy. LLMs such as InstructGPT (Ouyang et al., 2022), Chinchilla (Hoffmann et al., 2022), OPT (Zhang et al., 2022), GPT-4 (OpenAI, 2023), PaLM 2 (Anil et al., 2023) and LLaMA 2(Touvron et al., 2023b) are developed intensively and commercialized.

Regular finetuning adapts a generally pretrained model to a task-specific data distribution, which will incur another slightly different copy of the pretrained parameter sets. However, the rapidly increasing scale of LLMs makes it impractical to store and load several versions of these model finetuned for multiple users and tasks. As a light alternative, parameter-efficient finetuning methods only introduce or retrain a negligible portion of pretrained parameters, yielding a high degree of parameter sharing while preserving competitive performance as fully finetuning. Houlsby et al. (2019) inserts new adapter modules between layers of a frozen pre-trained model, but extends the depth of the original model, thus incurring more time latency. Lester et al. (2021) concatenates learnable prompt with input and feed this longer sequence into the frozen network. However, this method reduces the model's usable sequence length, needs much more computation, and is empirically verified to be sensitive to the prompt initialisation. Li & Liang (2021) attaches prefixed tokens to the K and V sequences of the transformer layer and avoids the first drawback, but still suffers the later two problems and brings more memory footprint. Zaken et al. (2021) proposes to only finetune the existing biases without introducing any new parameters and thus avoids all the above problems, but its capacity is so limited that it only shows inferior performance. In contrast, Hu et al. (2021) imposes a low-rank constraint on the weight updates and these new parameters can be merged with the pretrained weights during inference, resulting no increased time latency.

At the training stage, dropout randomly deactivates each neuron with a certain probability, while keeps all the neurons active in the reference phase. The operation can be regarded as injecting noise like a data augmentation method, preventing co-adaptation of neurons, or ensembling exponentially many sub-networks, and has been verified as a extremely effective regulariser to improve the generalization ability of multiple popular applications and model architectures, including CNNs, RNNs, Transformers and GNNs. Specifically for Transformer models, Zehui et al. (2019) proposes the first variant for fully-connected self-attention layer, dropping the attention weights to prevent different contextualized tokens from co-adaption. Instead of dropping the attention layer, Chen et al. (2021) introduces contiguous span-style deactivations to hidden representations produced by the linear layer following each attention layer. Recently, Li et al. (2023) proposes to drop key units before the softmax layer to keep probability features of attention weights. But it only focuses on computer vision tasks, while totally neglecting the NLP community that emphasizes on semantics and linguistic information. And we closes this gap with extensive experiments, verifying its superior performance on NLP tasks and some different conclusions. Wu et al. (2021) minimizes the bidirectional Kullback-Leibler (KL) divergence between the two output distributions of sub models sampled by two different forward passes with dropout. Shen et al. (2020) applies Jensen-Shannon Divergence loss to enforce consistent representations between outputs with and without dropout and thus narrows the gap between model training and inference stages.

## B   MATHEMATICAL PROOFS

We prove the mathematical equivalence of $w'_u$ for DropKey and DropAttention as follows:

$$
\begin{aligned}
\frac{\exp(g_u)}{\sum_{i=0}^{l-1}\exp(g_i)} \cdot \frac{1}{\sum_{i=0}^{l-1}\overline{w}_i} &= \frac{\exp(g_u)}{\sum_{i=0}^{l-1}\exp(g_i)} \cdot \frac{1}{1-w_m} \\
&= \frac{\exp(g_u)}{\sum_{i=0}^{l-1}\exp(g_i)} \cdot \frac{1}{1-\frac{\exp(g_m)}{\sum_{i=0}^{l-1}\exp(g_i)}} \\
&= \frac{\exp(g_u)}{\sum_{i=0}^{l-1}\exp(g_i) - \exp(g_m)} \\
&= \frac{\exp(g_u)}{\sum_{i=0,\neq m}^{l-1}\exp(g_i)} \\
&= \frac{\exp(g'_u)}{\sum_{i=0}^{l-1}\exp(g'_i)}
\end{aligned}
\tag{14}
$$

The proportional relationship of $\frac{\partial w'_u}{\partial g_u}$ between DropKey and DropAttention can be derived with the following equation:

$$
\begin{aligned}
\frac{(\frac{\partial w'_u}{\partial g_u})^{\text{DropKey}}}{(\frac{\partial w'_u}{\partial g_u})^{\text{DropAttention}}} &= \frac{\exp(g_u) \cdot (\sum_{i=0,\neq m}^{l-1}\exp(g_i) - \exp(g_u))}{(\sum_{i=0,\neq m}^{l-1}\exp(g_i))^2} \cdot \frac{\sum_{i=0}^{l-1}\exp(g_i) \cdot \sum_{i=0,\neq m}^{l-1}\exp(g_i)}{\exp(g_u) \cdot \sum_{i=0,\neq u}^{l-1}\exp(g_i)} \\
&= \frac{\sum_{i=0,\neq m}^{l-1}\exp(g_i) - \exp(g_u)}{\sum_{i=0,\neq m}^{l-1}\exp(g_i)} \cdot \frac{\sum_{i=0}^{l-1}\exp(g_i)}{\sum_{i=0,\neq u}^{l-1}\exp(g_i)} \\
&= \frac{1 - \frac{\exp(g_u)}{\sum_{i=0,\neq m}^{l-1}\exp(g_i)}}{1 - \frac{\exp(g_u)}{\sum_{i=0}^{l-1}\exp(g_i)}}
\end{aligned}
\tag{15}
$$

If $k$ is denoted as the result of Eq. 15, we have

$$
\begin{aligned}
k &< \frac{1 - \frac{\exp(g_u)}{\sum_{i=0,\neq m}^{l-1}\exp(g_i)+\exp(g_m)}}{1 - \frac{\exp(g_u)}{\sum_{i=0}^{l-1}\exp(g_i)}} \\
&= 1
\end{aligned}
\tag{16}
$$

## C   DATASET DETAILS

For NLU tasks, (i) Stanford Sentiment Treebank (**SST-2**) (Socher et al., 2013) is an English sentiment classification benchmark for a single sentence task, predicting whether the sentiment of movie reviews is positive or not. (ii) Recognizing Textual Entailment (**RTE**) (Wang et al., 2018) represents an inference task that predicts the entailment relation between two sentences. (iii) Microsoft Research Paraphrase Corpus (**MRPC**) (Dolan & Brockett, 2005) predicts the semantic equivalence between two sentences, while (iv) Semantic Textual Similarity Benchmark (**STS-B**) (Cer et al., 2017) predicts the similarity between two sentences. The later two tasks are involved with comparing and assessing the similarity and paraphrasing of two sentences. It is worth noting that, compared to the other classification tasks, STS-B performs a regression task and thus encompasses a broad range of tasks, enhancing the generalizability of our conclusions. Besides, additional experiments are further conducted to verify the validation of our analysis on (v) Corpus of Linguistic Acceptability (**CoLA**) (Warstadt et al., 2018), which aims to predict whether a sentence is linguistically acceptable or not, and (vi) Question Natural Language Inference (**QNLI**) (Rajpurkar et al., 2018), which predicts whether a sentence is the answer to a given question. For NLG tasks, we focus on (vii) **E2E** NLG Challenge (Novikova et al., 2017) and (viii) **WebNLG** (Gardent et al., 2017). The

former consists of sets of slot-value pairs along with multiple corresponding natural language references in the restaurant domain, while the later is a dataset where models generate corresponding description in form of natural language text given a sequence of SUBJECT-PROPERTY-OBJECT triples.

As for the evaluation metrics, we report the Pearson correlation for STS-B, Matthew's correlation for CoLA, and accuracy for other NLU datasets. For NLG tasks, BLEU, NIST, METEOR, ROUGE-L and CIDEr are used on the E2E NLG Challenge dataset, while BLEU, METEOR and TER are evaluated separately for "Unseen", "Seen" and "All" categories in the test set of the WebNLG dataset.

## D  HYPERPARAMETER CONFIGURATION

Table 6: The hyperparameters for RoBERTa-large and LLaMA-7B with LoRA on NLU datasets.

| Model | RoBERTa-large | | | | | | LLaMA-7B | |
|---|---|---|---|---|---|---|---|---|
| Dataset | RTE | MRPC | STS-B | SST-2 | CoLA | QNLI | RTE | MRPC |
| Optimizer | AdamW | | | | | | AdamW | |
| Weight Decay | 0.1 | | | | | | 0.1 | |
| Warmup Ratio | 0.06 | | | | | | 0.06 | |
| LR Schedule | Linear | | | | | | Linear | |
| Learning Rate | 4E-4 | 3E-4 | 3E-4 | 4E-4 | 2E-4 | 2E-4 | 4E-4 | 5E-4 |
| Epoch | 30 | 30 | 10 | 10 | 40 | 10 | 10 | 8 |
| Batch Size | 64 | 32 | 32 | 64 | 32 | 32 | 64 | 32 |
| Mac Seq. Len. | 512 | 512 | 128 | 512 | 128 | 512 | 512 | 512 |
| LoRA Rank | $r_q = r_v = 8$ | | | | | | $r_q = r_v = 8$ | |
| LoRA Scalar | 16 | | | | | | 16 | |

As shown in Table 6 and 7, we mainly follow the setup of LoRA (Hu et al., 2021) with as minimal changes as possible. However, based on our pre-experiments, significant fluctuations of the results are observed when models are trained with the original epochs, even if only random seeds change. Therefore, we increase the number of training epochs for more steady results. We also use the regular initialization instead of the MNLI checkpoint to initialize the LoRA modules. Different from RoBERTa-large and GPT2-Medium, we employ FP16 mixed precision training for LLaMA-7B to reduce the memory requirement.

For the specific parameters in our experiments, we disable dropout in baselines and iterate all available dropout rate from {0.01, 0.02, 0.05, 0.1, 0.15, 0.2} for various dropout methods, which is expanded with {0.25, 0.3} for clearer trend of performance in RTE dataset. To the best of our knowledge, neither of HiddenCut, DropKey and DropAttention implements experiments with a casual decoder-only transformer model before. Based on our empirical observation, applying any of these methods can only produce limited improvement or even deteriorate the performance on both NLU and NLG tasks, and the

Table 7: The hyperparameters for GPT2-Medium with LoRA on NLG datasets.

| Dataset | E2E | WebNLG |
|---|---|---|
| Training | | |
| Optimizer | AdamW | |
| Weight Decay | 0.01 | |
| Warmup Step | 500 | |
| LR Schedule | Linear | |
| Learning Rate | 2E-4 | |
| Epoch | 5 | |
| Batch Size | 8 | |
| Label Smooth | 0.1 | |
| LoRA Rank | $r_q = r_v = 4$ | |
| LoRA Scalar | 32 | |
| Inference | | |
| Beam Size | 10 | |
| Length Penalty | 0.9 | 0.8 |
| No Repeat N-Gram Size | 4 | |
| Repetition Penalty | 1.0 | |

results are extremely sensitive to the dropout rate. This phenomenon might be caused by fragile shallow forward process. In other words, noise introduced by dropout methods can be amplified with the propagation and diminish the benefits brought by dropout. Hence, we only introduce the dropping in the latter half of layers in decoder-only models and the apparent performance improve-

ment emerges again. Besides, our pre-experiments demonstrate that a weight between 0.01 and 10 for KL and JS loss generally yields the best results. Therefore, we iterate the weight within {0.01, 0.02, 0.05, 0.1, 0.2, 0.5, 1, 2, 5, 10}. All experiments are repeated 5 times to calculate the median values on NLU tasks, while the average values of three runs is reported for NLG tasks.

# E  LOSS LANDSCAPE VISUALIZATION

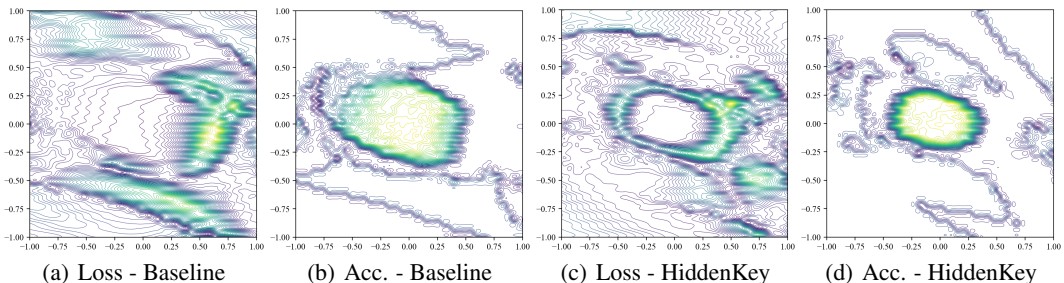

|  (a) Loss - Baseline | (b) Acc. - Baseline | (c) Loss - HiddenKey | (d) Acc. - HiddenKey |

Figure 6: 2D contours of loss and accuracy with respect to perturbed low-rank matrices in LoRA of baseline and HiddenKey.

Following Li et al. (2018), we also visualize the loss landscape to understand the parameter sensitivity and model generalization in PEFT scenarios. We utilize the filter normalization, introduced by Li et al. (2018), to correct perturbations by normalizing them based on the scale of parameters filter-wisely. In the original paper, based on the sensitivity of loss to model parameters, side-by-side comparisons of different minima can be enabled, and a flatter landscape is indicative of better generalization. Here we perturb low-rank matrices in LoRA along two random directions and visualize the loss and accuracy landscapes in Figure 6. However, the results clearly contradict this empirical notion. Compared with the landscapes of the baseline, those of HiddenKey exhibit rougher loss and accuracy surfaces, but significantly better performance, which might be attributed to the enhanced robustness of representations brought about by dropout. Based on the fact that loss is obtained by combining the model's structure, parameters and inputs, this contradiction suggests that the generalization and robustness of a model may involve loss sensitivity (flatness of the loss landscape) to both parameters and inputs (more broadly, input embeddings and hidden representations). One-sided analysis can provide some but not sufficient insights. Hence, building loss surface against parameters and representations simultaneously will further establish the relationship between loss landscape and model generalization. Although this is not the focus of this paper and left for future research, it may shed light on the future research direction of loss landscape visualization.

