# OpenReview forum: "HiddenKey: Parameter-Efficient FineTuning Meets Dropout under a Unified Framework"
_ICLR.cc/2024/Conference — Submitted to ICLR 2024_

### Official Review · Reviewer_jBFS · 2023-10-28

**Soundness:** 2 fair
**Presentation:** 2 fair
**Contribution:** 2 fair
**Rating:** 3
**Confidence:** 3

**Summary:**

The authors motivate their work on the gap between parameter-efficient finetuning and dropout regularization. They introduce a framework to unify PEFT and dropout called HiddenKey and demonstrate performance improvements with lower finetuning cost on two tasks.

**Strengths:**

- Mathematical comparison on how HiddenKey differs from DropKey and DropAttention was given.
- The authors have evaluated their proposed method on two holistic benchmarks, although as a minor point, the number of iterations to compute the standard deviations was not indicated.

**Weaknesses:**

- incremental improvement: In Table 1, the HiddenKey method with KL achieves the best score of 92.01 on average, but this is not a significant improvement (< 1%) from either DropKey or HiddenCut methods. Similarly, HiddenKey only demonstrated marginal improvements in Table 2 to 5, with the highest improvement in Table 2 CoLA (1.95%). I would not claim performance superiority given these results.

- lack of analysis: While the paper describes the intuition of how HiddenKey differs from DropKey, DropAttention, HiddenCut, it does not give any analysis to describe the difference in empirical experiments e.g. what features were learned or how the computations are empirically different. Minimally, an ablation study is necessary to elucidate which component of HiddenKey is required for the marginal performance gains.

- unsupported claim for faster convergence: The authors claimed that "HiddenKey outperforms baseline with shorter finetuning process". When training a model, the training loss is usually used as the metric to determine when model training should be stopped. In Fig.5, at the point of the black vertical line, the baseline model has clearly plateaued while HiddenKey is still decreasing. To reiterate, if i were to use the HiddenKey method, I would take reference from the plateauing of the training loss and not the accuracy, which in this case is significantly longer than the baseline method.

**Questions:**

- Fig 4 seems that DropKey method is more stable and contributes to a better performance than hidden method with increasing dropout rate?
- Is Fig 5 indicating training or test loss? A decrease in training loss does not indicate overfitting, as hypothesized by the authors.

---

> ### Author Response · Authors · 2023-11-16
> **Reply to Reviewer jBFS**
>
> 1. Considering that GLUE is a mature benchmark where the adoption of various techniques has established very strong baselines and left little space for further improvement, the consistent improvements over such strong baselines on both NLU and NLG tasks are still significant. Especially, HiddenKey improves the BLUE by 1.26 and NIST from 8.615 to 8.765 on the E2E dataset. Besides, we conduct repeated experiments to ensure robust conclusions and insights.
> 2. We analyze the integration of HiddenKey step by step, of which the process can be regarded as the ablation study. Hence, each component of HiddenKey has been verified to be useful.  Besides, we provide the formulation and comparison of various methods to demonstrate the computation.
> 3. Essentially, HiddenKey provides an option for users. If resources are constrained, HiddenKey can provide better performance with the same iteration number. As you said, if the longer finetuning process is allowed, HiddenKey can improve the performance further. In both cases, HiddenKey provides extra benefits.
> 4. QA
>     1. In Figure 4, DropKey does perform better and more stably than HiddenCut. As shown in Table 1, HiddenKey, combining HiddenCut, DropKey, and KL loss, provides the best performance.
>     2. The dashed and solid lines mean the training loss and test acc, respectively. The decreased acc with lower training loss in one trail means the occurrence of overfitting, while lower training loss does not mean better acc for different trails shown in Figure 5.

---

> > ### Comment · Reviewer_jBFS · 2023-11-21
> >
> > I thank the authors for their rebuttal, however, I find the response insufficient to motivate a change in my initial scores.
> > 1. Again, these are incremental performance and I would not claim performance superiority.
> > 2. Ablation studies of different components and comparison of various methods do not clarify what computation HiddenKey is actually performing. Network analysis to determine how HiddenKey improves performance or robustness will give better insights of what the algorithm is doing.
> > 3. What do you mean by "better performance"? Faster decrease in training loss or faster increase in evaluation accuracy? If it is the former, I would say the baseline model has better performance than HiddenKey. If it is the latter, the difference between HiddenKey and Baseline model is negligible. The only "better performance" I see is that HiddenKey achieves higher evaluation accuracy after 0.4 steps. Hence, I am not convinced HiddenKey provides an "option" for users.

---

### Official Review · Reviewer_4cjp · 2023-10-29

**Soundness:** 2 fair
**Presentation:** 2 fair
**Contribution:** 2 fair
**Rating:** 5
**Confidence:** 4

**Summary:**

This paper investigates three Transformer-specific dropout methods and their performances under PEFT scenarios, and proposes a new dropout scheme named HiddenKey to integrate the best of all the three methods.

**Strengths:**

1. A novel aspect: This paper investigates the dropout mechanism in PEFT methods, which is kind of novel to me.

2. Sufficient experiments: This paper conducts experiments on various tasks and datasets to validate the effectiveness of HiddenKey.

**Weaknesses:**

1.	Writing:
- The Introduction conclusively describes the phenomenon of overfitting without dropout in PEFT. After reading the Intro, I naturally associate this paper with “Dropout neurons in LoRA” or something like this. However, in Sec. 2 and 3, when formulating problems and methodology, there is nothing about PEFT, which makes me really confused about what this paper is really about.
- The contribution of this paper is not summarized, leading to more confusions.
- The results that validates the insight (dropout helps overfitting) should be a pivot experiment arranged before the method. Otherwise, the logic may fall in and jump out rapidly, and readers may lose concentration of the conclusion that the results intend to present.
2.  Investigated Methods:
When talking about dropout methods, it is so limited to only talk about DropKey, DropAttn and HiddenCut methods. At least, the very basic Dropout or DropConnect should be investigated. This paper also mentions dropout in so many aspects: Neuron-wise, input-wise, and also attention-wise. There are so many works concerning those aspects, and a comprehensive analysis can be really complicated. However, there is no systematic analysis concerning different types of dropouts. I suggest to focus on a more specific aspect (e.g., attention-wise). Otherwise, a case-by-case analysis in this paper should not be concluded with such a big title - “PEFT meets Dropout”.
3.  Theoretical Analysis:
When talking about preventing overfitting, some analyses about the generalization error bound is expected. However, this paper only conducts some gradient calculation to present “proofs” about reducing gradient noises. However, it is unclear what gradient noises are. There is also no formal definition or theories that can defines or support the noise reductions and lower error bounds of your methods.
4.  Novelty of the Proposed Method:
The core method is simply a mixture of three mentioned dropout schemes. I do not think this is novel enough for ICLR.

**Questions:**

I have no questions. The detailed suggestions please see Weaknesses.

---

> ### Author Response · Authors · 2023-11-16
> **Reply to Reviewer 4cjp**
>
> 1. We will refine our introduction part to make it more logical and clear. Our paper focuses on the empirical analysis of the effect of various transformer-specific Dropout methods on the PEFT performance. With this purpose, the main contribution is to create a framework to unify the design of different Dropout methods and compare their effectiveness in the context of PEFT. We conduct comprehensive experiments to evaluate the performance of nine different Dropout patterns and empirically conclude the best combination - HiddenKey, which is a recommended practice to alleviate the overfitting in the context of PEFT. Besides, we derive some inspiring insights and mathematical co-relation (the similarity in gradient flow of DropKey and DropAttn.) between different Dropout methods.
> 2. For the selection of Dropout Methods, we mainly focus on those specially designed for Transformer architecture. There are so many methods actually. As for the vanilla dropout, element-wise methods in our paper can be regarded as the basic Dropout, which will be explicitly mentioned in our paper.
> 3. Since the Method section focuses on comparing and unifying different dropout methods, we mathematically analyze the relationship between DropKey and DropAttn. For one iteration, it is hard to gain the clean and noisy gradient simultaneously, disentangle the clean and noisy part due to the complicated back-propagation, and determine the impact on performance. However, we can use detach() to stop the gradient noise to verify its existence.
> 4. Please refer to the first point for our detailed contribution.

---

> ### Comment · Reviewer_4cjp · 2023-11-23
>
> Thanks for the response, I would like to increase my score to 5, which is still below the accept threshold, as I prefer some analyses about the generalization error bound rather than some gradient calculation for a sufficient justification.

---

### Official Review · Reviewer_iCRS · 2023-10-31

**Soundness:** 2 fair
**Presentation:** 1 poor
**Contribution:** 2 fair
**Rating:** 5
**Confidence:** 4

**Summary:**

The authors propose a unified framework for various transformer architecture specific dropout variants (including DropKey, DropAttention and HiddenCut). Then, the authors propose a new dropout scheme called HiddenKey which combines DropKey dropout position and R-drop idea by adding a KL divergence loss between dropout applied and non-applied outputs. The authors use this proposed dropout scheme to LoRA based PEFT setting and show that LoRA training is overfitting and can be improved by HiddenKey technique. The authors show that the proposed approach can outperform LoRA fine-tuning on various NLU and NLG tasks. Finally, the authors present the training loss curves and downstream accuracy curves to show another evidence that LoRA fine-tuning is overfitting.

**Strengths:**

- The paper summarizes the existing transformer specific dropout methods together and proposes a unified framework to see them as variants with different dropout locations, dropout patterns, and recovery methods.
- The authors show that the proposed method: HiddenKey can (slightly) outperform vanilla LoRA fune-tuning

**Weaknesses:**

- There's missing analysis of training cost of HiddenKey. Especially, it uses double forward pass with KL and JS training, there must be memory consumption and iteration time implications.
- Overall, writing can be improved with more revisions. Even though the sentences are grammatically correct, some sentences look not natural. (e.g. the very first sentence in the abstract - if you could ask chatgpt like tools, it would tell you the sentence is not natural) Some more details are will be in the Questions section.
- Full fine-tuning baseline (not just LoRA) would help understanding the upper bound of the performance. It will be good to have at least RoBERTa model experiments.

**Questions:**

- The sentence at the end of page 1 (starting with However) and the next sentence are not naturally flow.
- At the end of introduction, DropKey, DropAttention and HiddenCut suddenly appear without any context. Please add some more explanation that those are from previous research and references for them.
- Figure 3 is Illustration of HiddenKey? Not DropKey?
- Is there a training cost analysis with KL divergence?
- In Figure 4, dropkey_column is HiddenKey? It will be useful if this is mentioned somewhere.
- In Table 1, superscripts in HiddenKey+KL seem strange? 1000.00, 5.00. And, why only HiddenKey- has two values? Shouldn't all HiddenKey variants have two values?
- In Figure 5, can you include the validation loss? It might behave similarly to the accuracy.

---

> ### Author Response · Authors · 2023-11-16
> **Reply to Reviewer iCRS**
>
> 1. We modify our introduction to make the logic flow more natural.
> 2. References of DropKey, DropAttention, and HiddenCut will be added in the introduction section.
> 3. Yes, Figure 3 is an illustration of HiddenKey. We will correct this typo.
> 4. Based on our experience, the introduction of KL loss does not significantly increase the training cost. There are two reasons. One is that the second branch does not need backward, which is much slower than forward. The other is that the second branch runs in the torch.no_grad() mode, which does not store the immediate states and thus is much faster than general forward. Besides, two branches can be computed simultaneously, and the increased training time could be ignorable. We will conduct comprehensive research and evaluate it to demonstrate the additional cost given by the KL loss, if any.
> 5. DropKey_column is the column-wise DropKey. HiddenKey is not included in the Figure 4. The figure is demonstrated for the illustration of the superiority of element-wise HiddenCut and Column-wise DropKey, indicating the rationale behind our proposed HiddenKey → combining the element-wise HiddenCut, the Column-wise DropKey, and KL loss to maximize the performance.
> 6. For HiddenKey-, two values of superscripts mean the dropout rates of element-wise HiddenCut and column-wise DropKey, respectively. For other variants, one value means the weight of KL loss for the optimal performance, and the dropout rates remain the same with HiddenKey-. As for the reason of 1000 (apparently larger than other values), this is caused by the STS-B dataset, which is a regression dataset, and the scale of KL loss is different from other classification tasks.
> 7. Sure, we will include it for your reference.
> 8. Responses to the weaknesses: The training cost of the proposed HiddenKey will be analyzed to illustrate the efficiency of our methods. Meanwhile, we will revise our writing to improve the logic flow and make it more natural and clear. Furthermore, introducing a full fine-tuned result as a baseline will be very helpful for illustrating the upper bound of our proposed method; we will also explore other PEFT methods to generate a more comprehensive framework of the effectiveness of different dropout methods in the context of PEFT.

---

> > ### Comment · Reviewer_iCRS · 2023-11-21
> >
> > Thanks for providing the answers to the review questions. However, I don't see any updated data or figures, yet. So, I keep my scores for now.

---

### Official Review · Reviewer_twKX · 2023-11-02

**Soundness:** 3 good
**Presentation:** 2 fair
**Contribution:** 2 fair
**Rating:** 5
**Confidence:** 3

**Summary:**

This paper first shows that LoRA also suffers from overfitting, then develops a unified framework to compare dropout methods in terms of methodology and performance. The authors finally propose HiddenKey, a dropout method with consistency regularization for LoRA by integrating existing methods. Extensive experiments demonstrate the superiority of HiddenKey in many NLU and NLG tasks.

**Strengths:**

1. The paper comprehensively compares existing dropout methods for transformer models in terms of their gradient behavior, compensation measure, providing valuable insights for method interpretation.
2. The paper provides benchmark results and empirical observations for applying dropout to LoRA, which are useful for future research.

**Weaknesses:**

1. The technical novelty for the proposed new method is limited. HiddenKey is an integration of existing dropout methods.
2. In Table 1, it is hard to evaluate different patterns based on the results. The values are close and have large variance. This makes the design choice of combining "elementwise HiddenCut" and "column-wise DropKey" a bit arbitrary.
3. In both NLU and NLG experiments, HiddenKey$^-$ does not exhibit a clear gain over the best baselines, and the improvement of HiddenKey is not very significant. It would be nice to include results on more NLG tasks like wikitext and cnn/dm to demonstrate.
4. (Minor) The authors only show llama's improvements on two small NLU tasks, which is insufficient to verify the method's effectiveness on generative LLMs. It would be nice to show llama's performance on some NLG tasks.
5. (Minor) All claims regarding PEFT methods are only verified on LoRA.

**Questions:**

See weakness.

**Details Of Ethics Concerns:**

No ethics concerns.

---

> ### Author Response · Authors · 2023-11-16
> **Reply to Reviewer twKX**
>
> 1. Our paper focuses on the empirical analysis of the effect of various transformer-specific Dropout methods on the PEFT performance. With this purpose, the main contribution is to create a framework to unify the design of different Dropout methods and compare their effectiveness in the context of PEFT. We conduct comprehensive experiments to evaluate the performance of nine different Dropout patterns and empirically conclude the best combination - HiddenKey, which is the analysis result instead of the only contribution. Besides, we derive some inspiring insights and mathematical co-relation (the similarity in gradient flow of DropKey and DropAttn.) among different Dropout methods.
>
> 2. The relatively large variance of our results is subjected to the task configuration (i.e. The performance of LoRA is sensitive to the initialization). To remedy this large variance, we repeat each NLU task five times and take the median value to reduce the randomness, given that the median value is more robust to outliers than the average value. In Table 1, we additionally show the variance for the necessity of repeated trials and readers’ reference. Compared with other published works (e.g., LoRA), our variance falls into a reasonable area. Figure 4. provides a visual illustration of our results, and the superiority of element-wise Hiddencut and column-wise DropKey over different task configurations is demonstrated better.
>
> 3. The gain given by HiddenKey- is minor before applying KL loss, but it still showcases a positive signal that the average/median performance increases. After applying the KL loss, the superiority of HiddenKey could be confirmed. Another factor contributing to the minor improvements is that the GLUE benchmark is widely adopted, and it is hard to gain large improvement on a strong baseline.
>
> 4. Considering the resource constraint, we are still trying to supplement the experiments on NLG tasks with LLaMA.
>
> 5. Among all PEFT methods, LoRA is one of the most popular methods widely studied in academia and industry. There is also some work to unify different PEFT methods, like “Towards a Unified View of Parameter-Efficient Transfer Learning.” We hope our analysis is also valid for other specific PEFT methods.

---

### Meta-Review · Area_Chair_1KK2 · 2023-12-05

**Metareview:**

The paper presents interesting empirical analysis on transformer-specific dropout methods with regard to PEFT, which gives a unified framework to see them as variants with different dropout locations, dropout patterns, and recovery methods. Additionally, authors propose a new dropout scheme called HiddenKey which combines DropKey dropout position and R-drop idea. Reviewers have raised several concerns regarding the incremental improvements, writing clarity, and lacking of analysis.

Strengths:

As agreed upon by the reviewers, include a comprehensive comparison of dropout methods tailored for transformer models, and some initial evidence that the proposed HiddenKey method, albeit with modest gains, can enhance training performance in certain PEFT scenarios.

Weaknesses:

1. There seems to be confusion about the main focus of the paper due to the introduction implying a direct connection between dropout and mitigating overfitting in PEFT that isn’t clearly supported throughout. The overall contributions need to be summarized more clearly.

2. Reviewers remain unconvinced about the novelty of the HiddenKey method, which integrates existing methods without significant innovation. They also call for a clearer analysis that explains the working mechanisms and contributions of HiddenKey components.

3. The paper’s structure and coherence in presenting the methodology and findings need refinement.

**Justification For Why Not Higher Score:**

All reviewers consistently vote for rejection.
While the submission shows potential, the reservations held by reviewers regarding its novelty, depth of analysis, and the impact of the proposed method reflect a need for further work. It's encouraging to see that the authors could take all the feedbacks into consideration for future submissions.

**Justification For Why Not Lower Score:**

N/A

---

### Decision · Program_Chairs · 2024-01-16

Reject